# Depression as Compared to Level of Physical Activity and Internet Addiction among Polish Physiotherapy Students during the COVID-19 Pandemic

**DOI:** 10.3390/ijerph181910072

**Published:** 2021-09-25

**Authors:** Anna Zalewska, Monika Gałczyk, Marek Sobolewski, Irena Białokoz-Kalinowska

**Affiliations:** 1Faculty of Health Sciences, Lomza State University of Applied Sciences, 14 Akademicka St., 18-400 Lomza, Poland; monikagalczyk@onet.eu (M.G.); ibialokozkalinowska@pwsip.edu.pl (I.B.-K.); 2Plant of Quantitative Methods, Rzeszow University of Technology, al. Powstancow Warszawy 12, 35-959 Rzeszow, Poland; mareksobol@poczta.onet.pl

**Keywords:** COVID-19, depression, Beck Depression Inventory, physical activity, IPAQ, internet addiction, Kimberly Young questionnaire, students

## Abstract

Objectives: The aim of the survey was to assess the level of depression correlated with physical activity and internet addiction among physiotherapy students of Polish universities during the COVID-19 pandemic. Methods: The survey was carried out via the internet among Polish physiotherapy students (141 respondents). The level of depression was assessed by the Beck Depression Inventory, physical activity by the International Physical Activity Questionnaire (IPAQ) in Polish and the level of internet addiction by the Kimberly Young Questionnaire. Results: It was found that 31% of those surveyed stated that they suffered from moderate or severe depression. The overwhelming majority of the respondents (92%) considered the level of their internet addiction as low. More physical activity had a positive effect on mental health. The overuse of the internet exacerbated depressive symptoms. Conclusions: The prevalence of depression observed in students is mainly related to distant learning systems. Therefore, regular physical activity is recommended as it is associated with a lower level of depression. It is also advisable to provide students with necessary psychological care. Excessive use of social media is not recommended to elevate mood as it makes depression symptoms worse.

## 1. Introduction

Health is a multidimensional concept that includes social, physical and mental aspects. The COVID-19 pandemic has been a global health issue for over 18 months, and it has led to many negative changes. The necessity to maintain social distance and use of personal protection equipment has made everyday activity difficult [1].

The period of social distancing has caused numerous changes in social and economic life. As a result of the threat posed by COVID-19, numerous educational institutions have decided to call off face-to-face classes and have adopted a remote style of teaching and learning. With this form of work and study, it is difficult to function without access to the internet [2]. Both distant learning and the transfer of a large number of activities to the online world have meant that many people started spending more time using computers, smartphones and other electronic devices [3]. It should be noted that universities in Poland were almost continuously closed from March 2020 to June 2021. Excessive and uncontrolled use of the internet may turn into addiction, which may result in mental health problems [4,5]. Addiction is defined as the body’s strong dependence on taking certain harmful substances or performing certain activities [6]. It is usually associated with the intake of substances that make an individual dependent on their effects. Nowadays, internet addiction disorder is observed in connection with certain compulsive behaviors, for instance, internet addiction [6]. Previous research has shown that regular physical activity is associated with a lower level of internet addiction among young people [7,8].

Physical activity is defined as the amount of exercise necessary for everybody to develop and maintain good health [9]. Moreover, it positively influences our psyche. Physical effort is considered a good method of dealing with mental fatigue: it improves mood, has a positive effect on proper functioning of the mind and reduces anxiety [10]. The closure of schools as well as the amount of time spent at home have reduced physical and social activity, which in turn may pose a negative effect on our mental state [11,12]. Research conducted during the COVID-19 epidemic shows a positive effect of regular physical activity on mental health [13,14,15].

Public gatherings were banned, movement of persons and access to public spaces were restricted and direct social interactions were limited. Restrictions in movement and the closure of parks, gyms, fitness clubs and swimming pools forced physically active people to change their lifestyle [16]. These people, forced to work or study online, had increased internet exposure time, potentially increasing their risk of developing internet addiction because, apart from online activities, there were no alternative forms of spending free time during lockdown. The pandemic situation caused people to grow into habits of excessive internet use which, with reduced physical activity, may contribute to the development of depression [17]. 

Even before the COVID-19 pandemic, attention was paid to the interaction between anxiety and depression [18]. Previous observations also show that the student population is at high risk of developing depression and is exposed to other mental health problems. These conclusions are especially true for medical students [19,20,21,22]. 

Healthcare professionals assisted by students of university-level medical faculties (e.g., in physiotherapy) are of importance during this pandemic. Increased workload, physical exhaustion, severe stress and inadequate personal protective equipment are experienced by healthcare workers. Despite their commitment to the continuous care of critically ill and dying patients, they are not able to save all of their care recipients. Not only are they exposed to different stressors, but they and their families are also at higher risk of being infected with COVID-19. 

The long-term effects of pandemics may be both emotional and functional. Considering a psychopathological point of view, the COVID-19 pandemic constitutes fairly new stressors for a team of mental health personnel. The impact of the current pandemic may lead to the outbreaks of post-traumatic stress disorder and depressive syndromes related to the functioning of the individual in COVID-19 reality. It also results in a shortage of medical staff due to burnout or mental exhaustion [23,24,25,26,27,28,29,30,31,32].

The aim of the study was to determine the level of depression and its correlation with physical activity and internet addiction among physiotherapy students of Polish universities during the COVID-19 pandemic.

## 2. Materials and Methods

### 2.1. Participants and Procedure

In November 2020, a cross-sectional online survey was sent to physiotherapy students from 4 universities in the eastern and central parts of Poland. The electronic questionnaire, along with information on the research and anonymity, as well as voluntary consent to participate, was sent to students on their official e-learning platforms in the form of a link to the Google form after 9 months of online learning. This method was found to be the safest and the fastest. From the received questionnaires, every fifth person in each year was randomly selected from the following universities in Poland: Medical University of Bialystok, Lomza State University of Applied Sciences, School of Medical Sciences in Bialystok and Medical University of Warsaw. Correctly completed questionnaires were obtained from 141 students (104 women and 37 men) aged 18–25. This population was considered homogeneous in terms of age and this factor was therefore not taken into account in further analyses. The study population was a representative sample for the population of physiotherapy students in Poland aged 18–25. The group of physiotherapy students was chosen considering not only the practical profile of their education, but also limited physical activity level they adopted during the period of social distancing. 

The inclusion criteria for the survey were: status of a physiotherapy student, age over 18, consent to participate in research and correct completion of the survey form.

The Senate Commission for Ethics in Scientific Research of the School of Medical Science in Bialystok, KB/18/2020.2021, approved the research and obtained the informed consent forms from the respondents. The respondents’ participation in the survey was voluntary and the course of the research was established on the Personal Data Protection Act of May 10, 2018 (Journal of Laws of 2018, item 1000), in accordance with the Regulation of the European Parliament and the Council (European Union) 2016/679 as of 27 April 2016, on the protection of individuals with regard to the personal data processing and on the free movement of such data, and the repeal of Directive 95/46/WE (General Data Protection Regulation). Informed consent for the research was obtained from the participants.

### 2.2. Methods of Assessing the Level of Depression, Physical Activity and Internet Addiction

#### 2.2.1. Beck Depression Inventory

The level of depression was assessed on the basis of the results obtained using the Beck Depression Inventory—a 21-item self-reporting questionnaire for evaluating the severity of symptoms (from 0 to 3 points) [33]. For each point, the respondent chooses one answer which, in his/her opinion, best describes his/her condition at the time. The summary measure can range from 0 to 63 points, where higher values indicate greater levels of depression [33]. Depending on the number of points obtained, the results are also classified into a 4-point adjective scale:0–11—no depression;12–26—mild depression;27–49—moderate depression;50–63—severe depression [33].

The value for Cronbach’s alpha reported in papers is >0.7 [34,35,36,37].

#### 2.2.2. International Physical Activity Questionnaire

Physical activity was assessed by applying the short version of the International Physical Activity Questionnaire (IPAQ) in Polish, which consisted of 7 questions concerning all types of daily physical activity [38]. Activity during work, activity at home and its surroundings as well as free time spent on other physical activities were taken into account. The respondents provided answers to the questions on the time spent on walking, sitting and during intense and moderate physical activity [38]. The questionnaire assessed the activities that last uninterruptedly for at least 10 min. All activities are reported in MET-min/week units (the product of the coefficient assigned to the activity and the number of days it was performed on during the week, as well as the duration in minutes per day) [38]. IPAQ is intended for people aged 15–69 [38]. The value for Cronbach’s alpha reported in papers is >0.7 [39,40,41].

#### 2.2.3. Kimberly Young Questionnaire

The Kimberly Young Questionnaire was used to assess the level of internet addiction [42]. It is a 20-item scale that measures the frequency of internet use. The respondents answer the questions on a 5-point scale. The total measure is between 20 and 100 points (higher values—greater degree of internet addiction) [42]. The respondents are assigned to three groups of internet addiction: low (20–49 points), medium (50–79 points) and high (80–100 points) [42]. The value for Cronbach’s alpha is >0.7 [43,44,45].

### 2.3. Statistical Methods

The following set of descriptive statistics was used in the study: mean with 95% confidence interval, median, standard deviation and skewness coefficient. The skewness coefficient clearly differs from 0, and for some measures its value is above 1, which means right-hand asymmetry. Due to the quite clear asymmetry of the analyzed psychometric measures, one should focus not only on the means but also (and even above all) on the medians in the compared groups. Therefore, non-parametric methods of statistical inference were selected in the form of the Kruskal–Wallis test, Mann–Whitney test and Spearman’s rank correlation coefficient. 

## 3. Results

### 3.1. General Description of the Level of Depression, Activity and Internet Addition

Of the participants in the survey, 73.8% were women and 26.2% were men. The majority (almost 70%) of the students stated they had no depression according to the BDI. 23% of the respondents answered that they suffered from moderate depression while 8% experienced severe depression. The overwhelming majority of the surveyed students (92.2%) considered their level of internet addiction as low.

A large asymmetry is typical for physical activity measures due to a small number of people with a very high level of physical activity and a significant number of people whose physical activity is 0. Therefore, a more reliable measure of the average level is the median, not the average (written up by in plus observations). The measure of intense activity (320 MET-minutes/week) is higher than the measure of walking (297 MET-minutes/week) and the measure of moderate activity has a much lower value (median equal to 160 MET-minutes/week). MET (metabolic equivalent of task)—a unit of activity; 1 MET corresponds to the energy consumption by the human body while sitting (for 1 min.). 

The IPAQ questionnaire was used to categorize participants into three groups according to the level of their physical activity. A low level of physical activity was declared by 57 people (40.4%), an average level of activity was found in 56 people (39.7%) and 28 people (19.9%) stated their level of physical activity was high.

The table below (Table 1) presents a summary statement of information on the distribution of psychometric measures in the studied population.

There are no significant gender-related differences in the level of depression (Table 2).

### 3.2. Sex and Place of Residence vs. Mental Health and Activity 

This section deals with a comparative analysis of level of depression, physical activity and internet addiction depending on the sex and place of residence of the respondents. The aim is to define the groups at special risk of negative mental health effects during the pandemic.

The following table shows the level of psychometric measures in relation to the place of residence (Table 3). The interpretation of the following results shows that the place of residence does not differentiate the addiction to the internet or the level of activity in a statistically significant way. The only exception is depression but only among men, although there are some interpretation difficulties here. Average cities “stand out” in minus because the level of depression among male students is significantly higher (median 12.5 points) than in the other two groups (median around 8).

### 3.3. Physical Activity and Internet Addition vs. Depression Level 

Research has been carried out to determine whether there is a link between the level of physical activity and internet addition.

In the women’s group, a higher level of activity has a positive impact on mental health, i.e., it reduces the Beck Depression Inventory values. In particular, the correlation between moderate effort and depression measure is important (*r*_S_ = −0.27; *p* = 0.0056 **). Among men, the correlations are stronger and statistically significant for all activity measures (the strongest correlation is between the total activity and Beck Depression Inventory: *r*_S_ = −0.43; *p* = 0.0086 **) (Table 4).

The correlations between moderate-intensity activity and total activity with the measure of depression are shown in graphical form in Figure 1.

The relationship between the frequency of internet use and the severity of depression was also examined. Such a correlation exists and is statistically significant regardless of the sex of the students. Quite a strong correlation is seen in the group of men (*r*_S_ = 0.57) and a slightly weaker one among women (*r*_S_ = 0.43) (Table 5).

The correlations between the intensity of internet use and the measure of depression are shown in graphical form in Figure 2.

## 4. Discussion 

The period of the pandemic was associated with isolation and social distancing and, consequently, limited interpersonal contact. Remote work and learning in addition to a ubiquitous sense of fear and uncertainty became part of everyday life [46,47,48,49,50,51].

Our own research has shown that only two thirds of the surveyed students had no depression. As many as 23% of those surveyed stated that they suffered from moderate depression and 8% from severe depression. This corresponds to other studies concerning the level of depression among students during the epidemic period. Similar studies conducted in the USA indicate a larger scale of the phenomenon; as many as 48.14% of students demonstrated a moderate to severe level of depression [52]. Other data from the USA shows the incidence of moderate–severe depression in 31.7% of freshmen [53]. Research among Greek students shows there was a 74.3% increase in scores for depression [54]. There is an increase in depressive symptoms in students during the lockdown, regardless of the history of mental disorders [55].

Life and science have “moved” to the virtual world. In the difficult times of isolation, social media [56] became a huge support. Research conducted among Polish medical students after 8 weeks of distance learning showed that, compared to when they were taking traditional classes, they were less active [57]. A survey conducted among Indian students indicates that apart from many problems related to anxiety and depression (42%), students had difficulties with poor internet connections and unfavorable home learning environments [57,58]. Similar conclusions were drawn by scientists from Pakistan, where the vast majority of students did not have permanent access to the internet, as a result of which the effects of online learning were not as expected [59]. A large number of social media tools also significantly improved the quality of crisis management in recent times [60]. During the pandemic, social media also became a potential threat (huge amounts of compressed information that could be overwhelming and a quick spread of fake news). It has been proven that the frequency of social media use increases in the event of natural disasters and other crises [61]. During the pandemic and lockdown, the overwhelming majority of the surveyed students considered their level of internet addiction as low. Research conducted in Indonesia shows that isolation did not affect internet addiction among the studied group of adults, clearly emphasizing the high incidence of this type of addiction [62]. During the pandemic, the level of internet addiction among Mexicans was estimated at 10.2 and 0.2% (moderate and severe) [62] and among Chinese at 4.3% (severe) [63].

The level of physical activity during the epidemic period among the studied group was low. These results confirm worrying global trends [13,64,65,66,67,68,69,70,71].

Previous long-term studies indicate that women are more likely to suffer from depression, which is related to sex characteristics [72]. A similar correlation was found in studies conducted during the pandemic [73,74]. In the group researched by the authors, sex did not differentiate the level of depression.

Research conducted in China indicated that students from urban areas expressed more anxiety and fear, but less sadness, than students living in rural areas [73]. In the studied group, the place of residence significantly differentiated depression among men. In medium-sized cities, the level of depression among male students was significantly higher.

Prolonged restriction of physical activity and isolation have a negative impact on depression as well as the feeling of anxiety and fear [75]. In the group covered by this piece of research, a higher level of physical activity had a positive effect on mental health. Numerous studies confirm that regular physical activity during the COVID-19 pandemic is associated with lower levels of anxiety and depression among the respondents [13,14,15].

Research has shown that a high level of physical activity is associated with a low level of depression. Accordingly, restricted physical activity in times of a pandemic crisis may produce highly negative health effects (the so-called “health debt”, because depression will obviously influence other aspects of our physical health).

Online activities are often mechanisms that help people cope with anxiety and depressed moods [76]. Paradoxically, abuse of these activities may increase the sense of anxiety and depression [77]. Our own research confirmed that the frequency of use of the internet during a pandemic influenced the severity of depression. This correlation was statistically significant regardless of the sex of the students.

The COVID-19 pandemic poses threats to both physical and mental health. During a crisis, people not only experience fear and anxiety but also change their current habits, such as physical activity [23]. In the available literature, physical activity is the most studied lifestyle factor. Apparently, the level of physical activity is most effectively assessed with the use of the IPAQ [78]. There is evidence from many meta-analyses that physical activity plays a protective role in reducing the risk of developing certain psychiatric disorders, and motor interventions can be considered to be effective supportive treatments for depression, anxiety, stress-related disorders and addiction [79].

## 5. Conclusions

Due to the long-lasting epidemic situation and uncomfortable preventive measures, the COVID-19 pandemic negatively influences higher education. There is a high level of depression among students. The long-term consequences of the pandemic are unknown and its effects are likely to persist for a longer period of time. Therefore, it is extremely important to provide students with the necessary psychological care and to implement supporting strategies. Regular physical activity is recommended as it is associated with a lower level of depression and anxiety. Excessive online activity is not recommended as a mood-enhancing tool, as the frequency of using the internet increases the level of depression.

The presented research highlights the fact that regular physical activity and careful use of the internet are an integral part of maintaining mental health during a pandemic. Another valuable tip for working on improving the mental resilience of students is motivating them to implement regular physical activity and to pursue their talents and passions outside the virtual environment. The academic community should participate in this process.

The strong points of this study included easy access to the study group, a low cost, not much time spent on the project and an ecological form of data collection. This study, however, had some limitations, such as a small size of the study group and the predominance of females as well as the subjectivity of the answers. In addition, the test subjects included only physiotherapy students. Therefore, national studies are needed in the future, targeting a larger sample of people. Subsequent surveys should also include representatives of different age or professional groups and be assessed using more objective methods.

## Figures and Tables

**Figure 1 ijerph-18-10072-f001:**
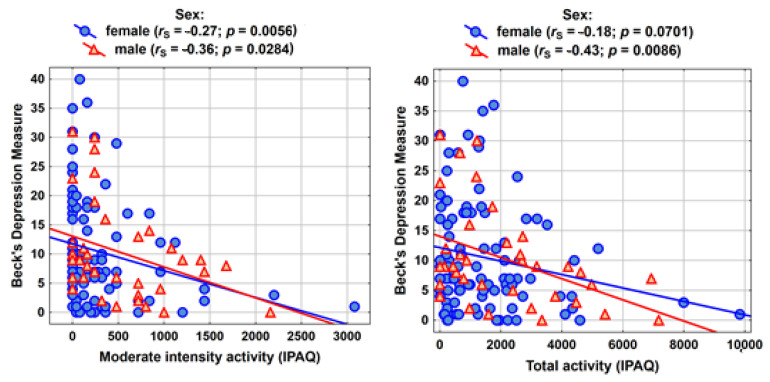
Correlations between moderate-intensity activity and total activity with the measure of depression. *r*_S_—Spearman correlation coefficient. *p* value—result of testing the significance of the Spearman correlation coefficient.

**Figure 2 ijerph-18-10072-f002:**
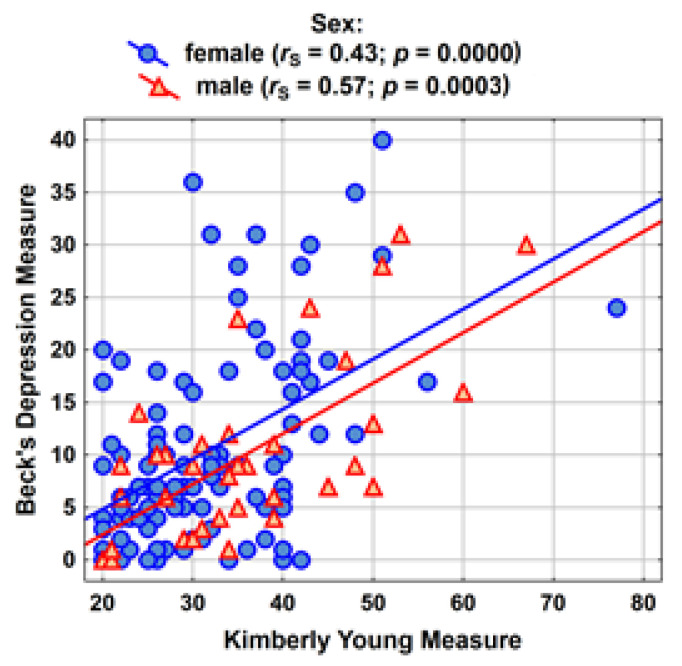
Correlations between intensity of internet use and the measure of depression.

**Table 1 ijerph-18-10072-t001:** Summary statement on the distribution of psychometric measures.

Psychometric Measures	Mean (95% c.i.)	Median	Std. dev.	Skewness	Kurtosis
Beck Depression Inventory (pts)	10.5 (9.0–11.9)	8	8.9	1.17	0.86
The measure of internet addiction (pts)	33.1 (31.4–34.7)	31	10.0	1.22	2.41
Intensive effort (MET-min./week)	778 (588–969)	320	1145	2.23	5.39
Moderate effort (MET-min./week)	338 (252–423)	160	513	2.41	7.18
Walking (MET-min./week)	501 (408–593)	297	555	1.49	2.24
Total effort (MET-min./week)	1617 (1324–1910)	1030	1760	1.82	4.19

**Table 2 ijerph-18-10072-t002:** Sex and measures of depression level.

Psychometric Measures	Sex				*p*
	Female		Male		
	Mean (95% c.i.)	Median	Mean (95% c.i.)	Median	
Beck Depression Inventory	10.6 (8.8–12.3)	7	10.1 (7.4–12.9)	9	0.9016

*p*-value was calculated using the Mann–Whitney test.

**Table 3 ijerph-18-10072-t003:** Level of psychometric measures in relation to the place of residence.

Psychometric Measures	Place of Residence						*p*
	Big City		Medium-Sized City		Small Town		
	Mean	Median	Mean	Median	Mean	Median	
Females							
Beck Depression Inventory	10.0	7	10.8	7	10.8	9	0.7657
Internet addiction	32.0	33	32.7	32	31.5	29	0.9471
Intensive effort	861	480	459	160	585	80	0.6083
Moderate effort	302	80	205	80	268	40	0.9827
Walking	520	396	442	215	577	297	0.5920
Total effort	1683	1272	1106	812	1430	693	0.5052
Males							
Beck Depression Inventory	9.9	7.5	15.0	12.5	7.1	8	0.0247
Internet addiction	37.0	35.5	39.0	37	33.4	33.5	0.3499
Intensive effort	820	640	848	680	1760	960	0.3209
Moderate effort	433	280	304	240	829	800	0.1426
Walking	338	264	511	248	473	396	0.6651
Total effort	1592	1174	1664	1205	3062	3346	0.1706

*p*—test probability value calculated using the Kruskal–Wallis test.

**Table 4 ijerph-18-10072-t004:** The level of physical activity and the occurrence of depression—values of Spearman’s rank correlation coefficients.

Activity Measures (IPAQ)	Sex	
	Female	Male
	Beck’s Depression Inventory	
Intensive effort	−0.17 (*p* = 0.0757)	−0.40 (*p* = 0.0142)
Moderate effort	−0.27 (*p* = 0.0056)	−0.36 (*p* = 0.0284)
Walking	−0.19 (*p* = 0.0594)	−0.37 (*p* = 0.0258)
Total effort	−0.18 (*p* = 0.0701)	−0.43 (*p* = 0.0086)

*p*-value—result of testing the significance of the Spearman’s correlation coefficient.

**Table 5 ijerph-18-10072-t005:** Computer addiction and depression severity—Spearman’s rank correlation coefficient.

Internet Addiction	Sex	
	Female	Male
	Beck’s Depression Measure	
Kimberly Young measure	0.43 (*p* = 0.0000)	0.57 (*p* = 0.0003)

*p*-value—result of testing the significance of the Spearman’s correlation coefficient.

## Data Availability

The data presented in this study are available on request from the authors.

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
