# Peer review of "Depression as Compared to Level of Physical Activity and Internet Addiction among Polish Physiotherapy Students during the COVID-19 Pandemic"

_ijerph, 2021, doi:10.3390/ijerph181910072_

Round 1

Reviewer 1 Report

Overall, the manuscript appears much improved, but still needs some work.

My suggestions:

ABSTRACT:

  1. Typically, I in internet is not capitalized since it is not a proper noun. Is there a reason for doing so?
  2. It may be better to just say Beck Depression inventory instead of 'Beck Scale'
  3. Refrain from starting a sentence with a number, e.g. 31%
  4. Online is a commonly accepted world. Hyphen between on and line is not necessary.
  5. There are too many grammatical errors throughout the paper. Consider using online or personal grammar reviewers for better presentation.

INTRODUCTION:

The newly added paragraphs, lines 68-80, should be revised to improve clarity of thought, and connection with previous statements, and the subsequent paragraph (aim of the study).

Author Response

Dear Reviewer,

Thank you for your very valuable comment on the article. We are very grateful to you for taking the time to assess our manuscript and for their positive and constructive comment.

Please find attached the revised version of the manuscript and responses.

We hope that edited version meets the standards of the journal and we are lookingforward to hearing from you.

We hope that applied the amendment will enable the publication of the manuscript in journal.

Yours sincerely,

Anna Zalewska

Reviewer 2 Report

At first, I would like to thank you for the effort that the authors put into this study.

My comments to the authors:

  • It would be essential to proofread the manuscript with a native English expert.
  • I believe the introduction should be improved since it's only given a general look to the readers. It would be ideal for providing more information about physical activity, depression, and internet addiction to the introduction instead of general knowledge of the last year.
  • Is not clear why line 69-76 is relevant on this study. This study is about polish physiotherapy students. The term "Healthcare professionals" is misleading.I recommend introducing the main target group in this section.
  • I recommend adding more studies on students' health behavior (including physical activity, addiction, and depression) during the COVID-19 pandemic
  • Include one subchapter before the measures (e.g., 2.2 Measure; 2.2.1. Beck depression inventory, etc.)
  • Numbering is missing on introduction and methods.
  • Please remove lines 115-116 and add Cronbach alpha to each measure.
  • All the measures need a reference
  • Please add reference how to calculate MET-min/week
  • A sentence like "Questions are closed and open-ended" is too general. Please be specific.
  • On the tables, there is some inconsistency with the numbers. Please use two decimals consistently everywhere.
  • Discussion should reflect more on the results. Please reconstruct after the corrections.
  • I believe 79 references is a lot. I suggest keeping what is essential for the close topic.

Author Response

(The authors gave the same response as above.)

Reviewer 3 Report

Thank you for your resubmission, as I said in my previous report it was almost there.

You have addressed most of my suggestions, and worked hard to significantly reshape the piece since the first draft.

On review of your cover letter theres a change that you claim to have addressed that has not been. Though you have said this is addressed in your cover letter , you still have not addressed the need for a reference lines 42-45 (originally lines 48-49). You do need a reference for your definition of addiction, as you say this is how it is defined, not how you are defining it so who says this? 

Other than that i have no further comment to make on this paper, I recommend it for publication and wish you well in your future endeavours.

Author Response

Dear Reviewer,

Thank you for your very valuable comment on the article. We are very grateful to you for taking the time to assess our manuscript and for their positive and constructive comment.

Please find attached the revised version of the manuscript and responses.

We hope that edited version meets the standards of the journal and we are lookingforward to hearing from you.

We hope that applied the amendment will enable the publication of the manuscript in journal.

Yours sincerely,

Anna Zalewska

This manuscript is a resubmission of an earlier submission. The following is a list of the peer review reports and author responses from that submission.

Round 1

Reviewer 1 Report

Introduction:

This section needs improvement and more literature review needs to be included. Some of the research studies mentioned in the discussion section can be included here to provide a better background for this research study. It is especially unclear from the introduction why the sample included specifically physiotherapy students. Authors must provide a rationale for studying this particular sample in this research.

Methods:

Did authors have any hypotheses while designing this study? If not, the aim of the study should be clearly delineated in this section.

Results:

The results section needs to be reorganized. It seems pretty haphazard at the moment and not easy to follow. Having a list of aims/hypotheses might also help with this organization. Psychometric properties of the measures used in the study are typically mentioned in the methods section. Is there a specific reason for them to mentioned in the results section?

Discussion and conclusions:

Discussion section seems disjointed and will benefit from including more transitional sentences and connecting literature review to the findings of the current research. What do authors mean by "more objective methods." (Ln 233). Another paragraph about the implications for this study could be included towards the end.

Author Response

Dear Reviewer,
Thank you for your very valuable comment on the article. We are very grateful to you for taking the time to assess our manuscript and for their positive and constructive comment.
Please find attached the revised version of the manuscript, together with responses to the reviewers.
We hope that edited version meets the standards of the journal. We hope that applied the amendment will enable the publication of the manuscript.

Yours sincerely,

Anna Zalewska

Reviewer 2 Report

The manuscript entitled "Depression in comparison with the level of physical activity and Internet addiction among Polish students during the COVID-19 pandemic" explores the relationship between depression, physical activity, and internet addiction during the pandemic. The study is interesting, and the topic is popular among researchers, but the manuscript needs to be improved.

I have the following comments to improve the manuscript:

Introduction

  • Introduction missing the literature review. It would be ideal that the authors provide previous studies on internet addiction and covid, physical activity and covid, depression, and covid, and their relationships in unusual times. Another aspect the authors should mention is the protective role of physical activity on measured factors (depression, internet addiction)

Material and Methods

  • I recommend renaming "Participants" to "Sample and Procedure" or Participants and Procedure
  • I recommend using a different subchapter for the three measures.
  • Report reliability (Cronbach alpha) of the depression and the internet addiction scale.
  • Add more information about IPAQ. "... consists of 7 questions concerning all types of daily physical activates" (line 98-99) - What kind of activities? What type of question was used? (Closed or open-ended)
  • Please report how MET-min/week was calculated

Results

  • I recommend using only one table to introduce the characteristic of the measured variables. In this table report, mean, SD, median skewness, and kurtosis
  • In table 2. please provide units or the minimum and maximum of the scales
  • Presentation of the numbers should be consistent: Please use two decimals on every result
  • It is not clear why only moderate intensity and total activity were correlated with depression. (Intensive, walking is missing)
  • Recommend presenting the results of internet addiction and depression in figures.

Discussion and conclusion

  • It is pleasurable that the authors put their results in international context, but I feel their results and conclusion does not get a recognition as it should.

Others

  • Extensive English proofread and style check needed

Author Response

(The authors gave the same response as above.)

Reviewer 3 Report

Thank you for your paper, this is a timely and important issue to explore that i believe, with significant restructuring in the way the information is presented here, will merit publication.  I have listed both major and minor issues below with suggestions of ways you could better restyle the article before final publication of this research.

There is a typo in the abstract -  "carried by" - this needs to be something like either 'conducted via' or 'carried out by' in order for this to make sense. There are occasional typing issues such as extra spaces or random typographical errors throughout - please be sure to proofread this carefully for such typing errors. This is not however a huge issue, just something to be fixed.

You state that "  Excessive and uncontrolled use of the Internet may turn into an addiction, which may result in mental health problems [6,7].  however internet addiction itself remains a debated topic, so it would be useful to nuance this with something that  acknowledges this- the same references would still stand as relevant however it is just important as it is a arguably a more nuanced modern addictive behaviour than some traditional models of addiction so i think it may be important to demonstrate that here. 

The article would also benefit from restyling or reorganisation of ideas as in some paragraphs you seem to jump between ideas - this can be a simple pitfall of co-authoring papers where ideas get muddled but for clarity separating them out would be beneficial: e.g.:  "Physical effort is considered a good method of dealing with mental fatigue, it improves mood, has a positive effect on the proper functioning of the mind and reduces anxiety [9]." and then within the same paragraph at the next sentence you jump straight to "Public gatherings were banned, movement of persons and access to public spaces were restricted..." which is two totally separate ideas.

The introduction is useful and relevant but very short and missing much of the detail that you have included in later sections - again some work on separating out some of the ideas here would enhance clarity, a restructure will also help identify where the content needs further development.

The methodology appears appropriate, the approaches undertaken make sense and appear sensible in line with the study. Again, more detail to explain some of your analytical decisions would be beneficial here.

Clarity surrounding how you decided the cut offs for the groupings of low/med/high would be beneficial here. How did you decide them - was it based on a (currently absent) citation for a paper that has established them as meaningful thresholds? Please clarify. 

I also wonder why you've categorised them at all rather than analysing the scores on a continuum as there is a risk of sensitivity loss when categorising when you originally had the full range of scores so you need to justify the decision more explicitly in order to demonstrate clearly to the reader that there is no data manipulation  to find a significant result.

The results and findings generally make sense and appear sensible though again there are issues of clarity in how they are presented. The expressions are mixed which impacts clarity - in the same sentence you use a textual expression of "two thirds" followed by precise statistics for other findings. Please provide the precise figures for each finding as it is unusual and risks appearing deceptive to make some findings more vague than others. "Two-thirds of the respondents did not feel depressed, 23% assessed that they suffered 121 from moderate depression, and 8% severe." The limitations surrounding participant sex numbers need to be acknowledged earlier, with a statement that acknowledges this limitation, but also explains the beneficial nature of your study aside from that.

A citation for the Beck inventory is needed, same for the other measures utilised as they are named but no citation is provided e.g. IPAQ, Kimberly Young.

Some of the content introduced in the initial section of the discussion would have been beneficial included in your introduction, perhaps use some of these ideas to enhance your introduction, which as i mentioned is very brief and could benefit from greater detail.

The discussion reads more like a list than an analytical discussion, and i actually think some of this would be better placed within the findings section to unpack the presented data rather than this listing and disjointed style currently presented. More attention here needs to be given to the specific data that you have found, and much of the summary of other research should be connected here, but i feel would be better placed in the introduction section to be cross referenced. There are important and interesting ideas here but ideally this would be better constructed into a better flowing and discursive narrative. At the moment it seems quite bullet point like which at times is detrimental to its readability, impacting the way the messages are conveyed. 

The strengths and limitations of the study are also dropped in as a list at the end of the discussion rather than included within the conclusion which may be more appropriate as it is part of your consideration of the strengths and/or limitations of the study which would usually feed into your recommendations for future research within the concluding section. 

The conclusions are also too short and don't really encapsulate what you found - this is more of a last paragraph of the conclusions/ closing statement than a conclusions section. The final line is an assertion that is unclear - recommended by who? wider literature or this research? please clarify such statements with either detail, citation or both.

I think this paper looks at an important topic and it is a timely and significant topic however in terms of the way the information is presented the paper would benefit from a significant restructure. with the development of  clearer and more connected discussion and conclusions sections, with some of the content currently interpreted in the discussion moving to the findings enhancing understanding and flow within this article. A restructure and restyle would really help the content here and if this is achieved i believe this will become a relevant and useful article that merits publication. 

Author Response

(The authors gave the same response as above.)

Round 2

Reviewer 1 Report

Title: consider including "physiotherapy" before students in the title, since that was the primary subject group, and inclusion criteria of the study.

Introduction: Please consider adding research studies in physiotherapy students, specially those pertaining to the level of stress and depression in this group, and if there are any studies that compare their level of stress to the general Polish population or other medical students.

Methods: Please state the source of funding for the study, and if the participants were compensated for their time in any way.

Results: Consider defining the sizes of the city category.

Discussion: It would be helpful to write the discussion bearing in mind that this is a correlation study. Confounding variables were neither mentioned nor measured in the population, which is also a pertinent limitation of the study. Since the point of the study is to show that 31% had depression, which is a considerably high number for this population, it would be helpful to mention prior research on what is the usual level of depression in physiotherapy or medical students.

Reviewer 2 Report

Dear Authors,

The manuscript improved a lot; however, I still have a few suggestions before publishing.

  • The aim of the study should be in the end of the introduction. Please move line 80-82 to end of the introduction. I also recommend using hypotheses as well.
  • "Methods for assessing the level of depression, physical activity and Internet addiction" (line 110): Please name it as "measures"
  • Line 110-111 is unnecessary, instead please provide the exact Cronbach alphas to the scales.

Reviewer 3 Report

Thank you for the resubmission of this piece, you have clearly worked hard on it and amended it significantly.

The language and presentation are greatly improved, the structure is significantly better and the clarity is much better.

The introduction now contains a much more appropriate amount of information, is well referenced and sets up the rest of the paper much better than the original.

Lines 48-49 do need a reference for your definition of addiction however, as you say this is how it is defined, not how you are defining it so who says this?

Lines 68-70 you claim that people have been "exposed to internet addiction" however what would be more accurate would be to say they have "had increased internet exposure time, potentially increasing their risk of developing internet addiction.

Lines 90-93 it is unusual, ethically, to name where you got your participants from. Totally fine if you do have consent to do this but might be worth a double check of your ethical approvals and anonymity protocols to be sure you have consent to publish this information, and have informed your participants as such.

Line 216 i believe you mean graphical "form" not graphical "from"

I would have still liked a further developed conclusion however the conclusion that you provide is much more fit for purpose

It is clear to me that you have taken all of the comments given by the reviewers on board and i am happy to recommend the publication of this paper, pending very minor amendments as detailed in this short follow up report. Well done.